# Determinants of EMNEs' Entry Mode Decision with Environmental Volatility Issues: A Review and Research Agenda

**Yameng Li [1], Ruosu Gao [1],* and Jingyi Wang [2]**

[1] Department of Research, International Engineering and Technology Institute, Denver, CO 80202, USA; Yolanda.li828@gmail.com

[2] School of Management and Governance, UNSW Sydney, Sydney 2052, Australia; jingyi.wang4@unsw.edu.au

\* Correspondence: ruosu1997@gmail.com

**Abstract:** Emerging market multinational enterprises (EMNEs) play a vital role in global economic development and usually adopt aggressive internationalization strategies. However, the volatile global environment has caused EMNEs to face various risks in their overseas expansion. To maximize the competitive advantages and achieve successful expansion, EMNEs should choose the most suitable foreign entry mode. Therefore, EMNEs need to understand what environmental factors affect their decision-making and how they influence the choice of entry modes, especially in a volatile environment. This review examines 44 selected journal articles from 1996 to June 2021 on the environmental volatility determinants of EMNEs' entry mode choice. The entry mode choice we examined is mainly wholly-owned subsidiary versus international joint venture. We categorized the environmental volatility determinants investigated in the literature we reviewed into country-level factors (such as cross-national distance) and industry-level factors (such as industry condition). The main contributions are: (1) the review reveals three research gaps in extant studies, which are lack of research on external environmental factors, lack of research on multinationals from less concerning emerging economies, and lack of research on small-to-medium (SMEs) enterprises. (2) Practically, the study highlights the importance of understanding external environmental factors for EMNEs to make the most suitable entry mode decisions.

**Keywords:** emerging multinational enterprise; entry mode; environmental volatility; systematic literature review; wholly-owned subsidiary; international joint venture

## 1. Introduction

Over the past two decades, emerging market multinational enterprises (EMNEs) have rapidly internationalized into both developed and developing markets (Pattnaik et al. 2020), contributing to the development of the global economy. Scholars are becoming increasingly interested in studying EMNEs' outward foreign direct investment (FDI). Some scholars focused on the relationship between EMNEs' international expansion and innovation capabilities (Hensmans and Liu 2018; Thakur-Wernz and Samant 2019). Sun et al. (2012) developed a comparative ownership advantage framework of EMNEs' FDI strategies. Liang et al. (2021) reviewed relevant research on EMNEs' strategic asset-seeking mergers and acquisitions. While Liu et al. (2021) provided a literature review on the co-evolution of EMNEs. Various influencing factors of entry mode have been studied, such as the speed to enter and access the new market (Gilroy and Lukas 2006), the existence of competitors in the host country (Wang 2009), the growth of the local industry and the availability of local experience (Makino and Neupert 2000).

The global environment is constantly changing, especially the ongoing trade war between China and the US since 2018 and the COVID-19 pandemic since early 2020, which increase risks to EMNEs' foreign investment. The volatile environment poses many risks

to EMNEs' outward FDI, including political, economic, and social risks. Therefore, it is important to study which environmental factors affect EMNEs' entry mode choice for outward FDI and how they influence decision-making. It will help EMNEs to have a more comprehensive understanding of the investment environment and then make a more efficient choice of internationalization entry mode.

This paper reviews the literature on the environmental volatility determinants of EMNEs' entry mode choice. We reviewed and analyzed 44 selected papers regarding the environmental determinants towards EMNEs' entry mode choice published in 19 journals from 1996 to June 2021. We summarized the environmental factors that influence EMNEs' entry mode decision-making at either country-level or industry-level discussed in the existing literature and classified them into detailed categories. We contribute to the literature in two aspects. First, we provide a holistic picture of the available discussion on the volatile environmental determinants of EMNEs' entry mode choice. Such an overview of research about the volatile environmental determinants of EMNEs' entry mode choice provides scholars with the state of knowledge in this specific research area. Second, we have identified the following three research gaps in the existing literature and provided future research directions to scholars. (1) Most papers focus on firm-level factors; thus, research on environmental volatility factors that affect EMNEs' entry mode choices is limited. (2) The existing research mainly focuses on multinationals from large emerging countries, such as China and India. More research on other small emerging countries is needed to understand EMNEs' internationalization strategy fully. (3) The existing studies emphasize large companies, such as listed companies, calling for more insights into small and medium-sized companies.

The remainder of the paper is organized as follows. In Section 2, we described the research method used for conducting the review. Then, we provided some descriptive statistics regarding bibliometric analysis in Section 3, followed by a content analysis in Section 4, in which we classified the environmental factors as country-level factors and industry-level factors for detailed analysis. Next, in Section 5, we proposed opportunities for future research. Finally, we concluded the paper in Section 6.

## 2. Methods

A systematic literature review (SLR) is an approach adopted to confirm, evaluate, and synthesize all relevant research regarding a particular field that has been done and recorded by scholars (Bhimani et al. 2019). SLR aggregates and processes information as a scientific way as possible and limits biases, thereby identifying knowledge or research gaps in the particular field and pointing out the directions for future research (Moraes et al. 2021). We followed all three stages to conduct the SLR introduced by Tranfield et al. (2003): planning the review, conducting the review, and reporting and disseminating findings.

### 2.1. Stage I: Planning the Review

In this stage, we delimited the review topic, determined the objective, and defined the review's scope. The topic of our review is the environmental determinants of the EMNEs' entry mode decision. In addition, the objective of our review is to synthesize all relevant extant literature in this field. Therefore, we have excluded literature focusing on developed MNEs. Moreover, most of the extant studies focus on equity ownership-related issues; thus, we excluded studies investigating non-equity ownership-related issues as well. Additionally, to define the appropriate temporal scope of our review, we have included all published articles on EMNE's entry mode after the seminal article of Pan (1996), which made the search timespan of our review as 1996 to June 2021.

### 2.2. Stage II: Conducting the Review

There are four steps in the conducting stage, which are summarized in Figure 1. In the first step, we reviewed and analyzed influential articles and generated several keywords in accordance with our review topic, such as "EMNE", "emerging economy", "entry mode",

"FDI", "equity ownership", etc. We then formed two groups of keywords in the second step. The first group was made up of words that describe the EMNEs. For example, "EMNE", "emerging economy MNE", "emerging market multinationals", "Chinese MNE", "BRIC MNE", "Indian MNE", "Latin American MNE", etc. The second group consisted of words regarding entry mode. For instance, "entry mode", "wholly owned-subsidiary", "FDI", "cross-border acquisition", "CBA", "IJV", "joint venture", "greenfield", etc.

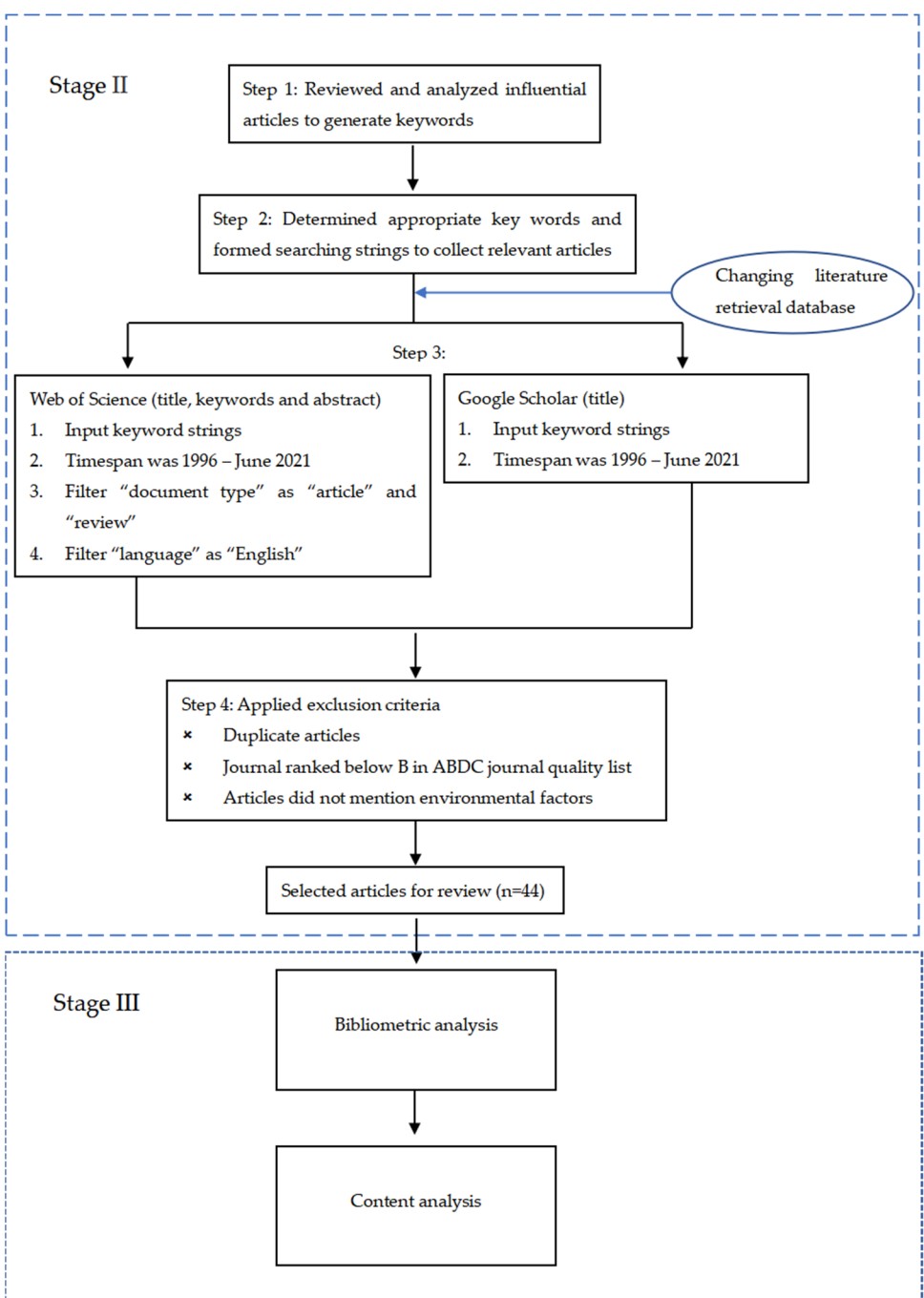

**Figure 1.** Stages II and III of the SLR method are conducted in this paper.

In the second step, we then selected keywords from both groups and used them together to form appropriate searching strings to aggregate relevant articles. By using the searching strings determined in the second step, we did a pair-wise keyword search (Chhabra et al. 2021) in the third step. We input the searching strings as "title, keywords,

and abstract of the article" in the Web of Science database. Other search criteria were set as follows: (1) search period: 1996–June 2021, (2) document type: "article" and "review", and (3) language: English. Meanwhile, we conducted a string search using the same pair of keywords in the "title" and the same search period in the Google Scholar database to supplement the article collection. In the last step, we applied further exclusion criteria to select relevant and appropriate papers. We first identified and removed duplicated articles and then only retained articles published in B, A and A* journals on the ABDC Journal Quality List. Further, we read and analyzed all the articles selected based on the above steps to identify if the studies involved environmental factors. The selected papers had been classified into two categories: (1) environmental factors have been taken into account of EMNE's entry mode decision making, (2) environmental factors have not been taken into account of EMNE's entry mode decision making. Since the objective of our review is to identify the environmental determinants of the EMNE's entry mode, we only retained the papers in the first category and excluded those without involving external environmental factors. As a result, there were 44 papers for our review.

### 2.3. Stage III: Reporting and Dissemination of Findings

This stage includes two main parts, which are "descriptive analysis" and "thematic analysis" (Tranfield et al. 2003). This view conducted a bibliography analysis for the descriptive analysis in Part 3 and content analysis for the thematic analysis in Part 4, respectively.

## 3. Bibliometric Analysis

### 3.1. Journal and Year Distribution

Our review consisted of 44 selected articles that came from 19 journals listed in Table 1. Among all 19 journals, five of which were identified as influential journals, including (1) Asian Pacific Journal of Management, (2) Journal of International Business Studies, (3) Journal of International Management, (4) Journal of World Business, and (5) Management International Review. As Figure 2 shows, most selected papers are from these five journals, which accounted for approximately 55%.

**Table 1.** Scholarly journals published articles on EMNE's entry mode decision-making involving volatile environmental factors.

| No. | Journal Name | Number of Relevant Articles |
|---|---|---|
| 1 | Asia Pacific Journal of Management | 4 |
| 2 | British Journal of Management | 1 |
| 3 | Corporate Governance—An International Review | 1 |
| 4 | Cross Cultural and Strategic Management | 2 |
| 5 | Emerging Markets Finance and Trade | 1 |
| 6 | Global Strategy Journal | 2 |
| 7 | International Business Review | 2 |
| 8 | International Journal of Emerging Markets | 1 |
| 9 | International Journal of Technology Management | 1 |
| 10 | International Marketing Review | 1 |
| 11 | Journal of Business Research | 3 |
| 12 | Journal of International Business Studies | 6 |
| 13 | Journal of International Management | 6 |
| 14 | Journal of World Business | 4 |
| 15 | Management Decision | 2 |
| 16 | Management International Review | 4 |
| 17 | Organization science | 1 |
| 18 | R & D Management | 1 |
| 19 | Thunderbird International Business Review | 1 |
|  | Total | 44 |

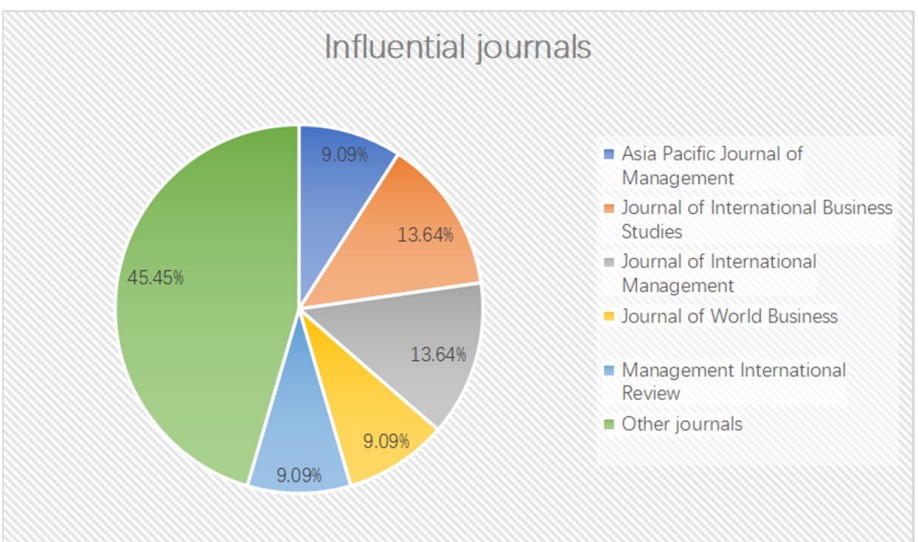

**Figure 2.** Influential scholarly journals on publishing articles on EMNE's entry mode decision-making involving volatile environmental factors.

Although the retrieval research period was from 1996 to June 2021, the research on EMNEs' entry mode decision-making involving environmental factors was limited until 2013. As shown in Figure 3, relevant research gyrated up from 2014. Further, 2017 was the year with the most publications. The total number of publications from 2014 to June 2021 was threefold the total number from 1996 to 2013, at 33 and 11, respectively.

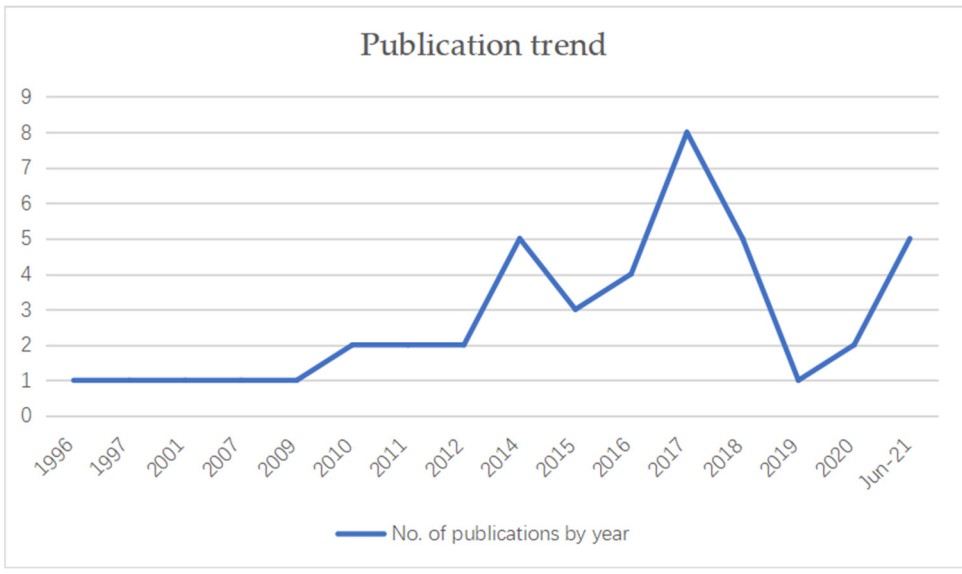

**Figure 3.** Numbers of selected articles published by year.

*3.2. Theories/Perspectives Used*

Of the 44 selected articles reviewed, more than fifteen theories or perspectives have been used. As Table 2 shows, chief among them are the institution theory/institution-based view (IBV) and the transaction cost economics (TCE), which account for approximately 37% and 20%, respectively. TCE has been used throughout the selected review period from 1996 to 2021, while institution theory/IBV has been used especially in the past 15 years. Following institution theory/IBV and TCE are the RBV/KBV/dynamic capabilities, which account for about 9% and have been used mostly in the articles published in 2017, 2018 and 2019. In addition, agency theory and bargaining power theory have been used for the

same times at four, while information asymmetry and OLI model/eclectic paradigm have been used both for three times. Other theories/perspectives, such as contingency theory, cultural/cultural distance theory, Uppsala model, etc., are less frequently used.

**Table 2.** Theories/perspectives of entry mode choice under volatile environments used in selected journal articles.

| Theories/Perspectives | Total Number | Percentage (%) |
|---|---|---|
| Institution theory/institution-based view (IBV) | 34 | 36.56 |
| Transaction cost economics (TCE) | 18 | 19.35 |
| RBV/KBV/dynamic capabilities | 8 | 8.60 |
| Agency theory | 4 | 4.30 |
| Bargaining power theory | 4 | 4.30 |
| Information asymmetry | 3 | 3.23 |
| OLI model/eclectic paradigm | 3 | 3.23 |
| Organizational capability theory | 2 | 2.15 |
| Resource dependence theory | 2 | 2.15 |
| Springboard perspective | 2 | 2.15 |
| Contingency theory | 1 | 1.08 |
| Cultural/cultural distance theory | 1 | 1.08 |
| Social network theory | 1 | 1.08 |
| Stakeholder perspective | 1 | 1.08 |
| Uppsala model | 1 | 1.08 |
| Others | 8 | 8.60 |
| Total | 93 | 100 |

### 3.3. Emerging Economies Have Been Studied

Figure 4 shows the emerging economies that have been studied in the selected articles and the number of their occurrences. The results reveal that China appeared the most with 27 times in all 44 articles. India, Russia, Brazil, and South Africa appeared 14, 11, 11 and 8 times. These five economies together happen to be the BRICS economies. Moreover, Turkey, Mexico and Southeast Asian economies (e.g., Malaysia, Indonesia and Thailand) are also essential economies being studied. Other economies, for example, United Arab Emirates (UAE), Central and Eastern Europe (CEE) and Latin America economies, have been relatively less studied.

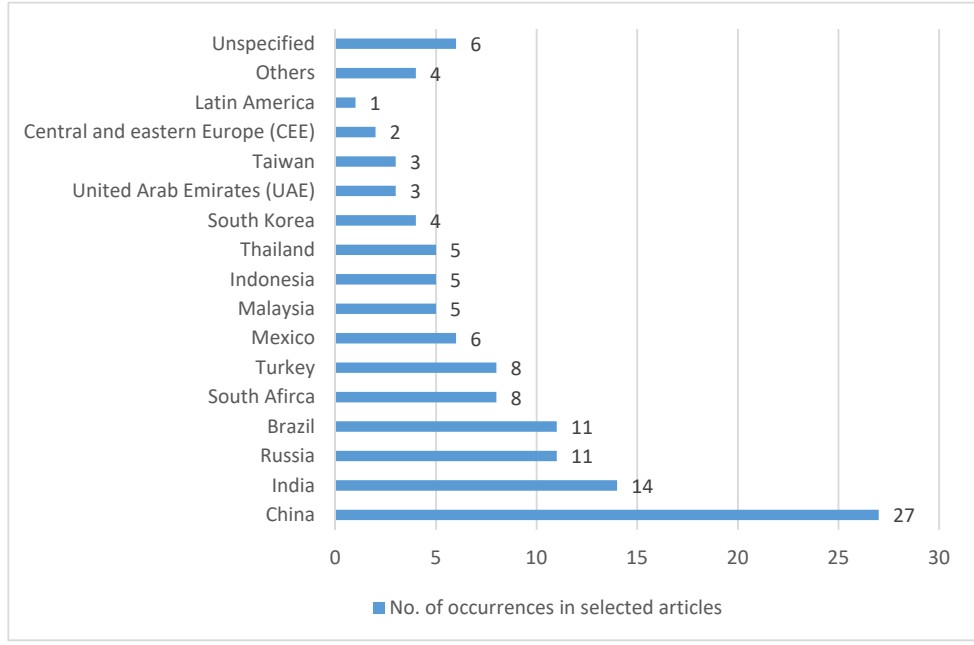

**Figure 4.** Numbers of occurrences of emerging economies in selected articles.

## 4. Content Analysis

To analyze how environmental volatility factors affect EMNEs' entry mode, we established a discussion model for 44 of the references from the country- and industry-level factors. Factors from the country-level include cross-national distance, cultural attributes, government-related factors, and historical and social factors; factors from the industry-level include industry conditions, financial indicators at the industry-level, industrial characteristics and trends, and technological level of domestic industries.

### 4.1. Country-Level Factors

Country-level factors discussed in the papers come from four dimensions: (1) cross-national distance; (2) cultural attributes; (3) government-related factors; and (4) historical and social factors (see Table 3).

**Table 3.** Country-level environmental volatility factors that affect EMNEs' entry mode choice.

| Independent Variables | Reference |
| --- | --- |
| *Cross-national distance* | |
| Institutional distance | (Wu et al. 2021; Malhotra et al. 2016; Liou et al. 2016; Kedia and Bilgili 2015; Ellis et al. 2018; Pinto et al. 2017; Kittilaksanawong 2017; Lahiri et al. 2014; Yang 2015; Liou et al. 2017a; Scalera et al. 2020) |
| Cognitive distance | (Ilhan-Nas et al. 2018; Liou et al. 2017b) |
| Normative distance | (Ilhan-Nas et al. 2018; Liou et al. 2017b) |
| Regulative distance | (Ilhan-Nas et al. 2018; Liou et al. 2017b) |
| Cultural distance | (Malhotra et al. 2011; Malhotra et al. 2016; Yang 2015; Pan 1996; Demirbag et al. 2009; Kim et al. 2020; Tseng and Lee 2010) |
| Sociocultural distance | (Mjoen and Tallman 1997) |
| Linguistic distance | (Dow et al. 2016; Demirbag et al. 2009) |
| Religious distance | (Dow et al. 2016) |
| Geographic distance | (Malhotra and Gaur 2014) |
| Psychic distance | (Chikhouni et al. 2017) |
| *Cultural attributes* | |
| Home country power distance | (Richards and Yang 2007) |
| Level of linguistic diversity in the host country | (Dow et al. 2016) |
| Level of religious diversity in the host country | (Dow et al. 2016) |
| *Government-related factors* | |
| Relationship between home and host countries | (Demirbag et al. 2010) |
| Host country's risk | (Richards and Yang 2007; Pan 1996; Demirbag et al. 2009; Demirbag et al. 2010; Moschieri et al. 2014; Lu et al. 2018; Tseng and Lee 2010; Bretas et al. 2021) |
| Host country's environmental dynamism, complexity, hostility | (Luo 2001) |
| Host country's digital media freedom | (Li et al. 2021) |
| Host country's partner state ownership | (Pan 1996) |
| Host country's market potential | (Nguyen and Binh 2021; Malhotra et al. 2011) |
| Host country's market orientation | (Luo 2001) |
| Host country's export orientation | (Luo 2001) |
| Host country's government effectiveness | (Lahiri 2017; Driffield et al. 2014) |
| Host country's corruption level | (Lahiri 2017; Driffield et al. 2014; Demirbag et al. 2010) |
| Host country's law and order | (Lahiri 2017; Driffield et al. 2014; Demirbag et al. 2010) |
| Host country's institutional environment | (Chen et al. 2017; Pan et al. 2014; Falaster et al. 2021) |
| Host country's legal requirements | (Mjoen and Tallman 1997) |
| Host country's strength of IP institutions | (Ahammad et al. 2018) |
| Host country's regulatory restrictions | (Cui and Jiang 2012) |
| Home country's government pressure | (Chung et al. 2016) |
| Home country's government support | (Pinto et al. 2017) |
| Home country's regulatory restrictions | (Cui and Jiang 2012) |
| Host country's factor market | (Chen et al. 2017) |
| Foreign origin and industry clusters | (Shen and Puig 2018) |
| *Historical and social factors* | |
| Colonial ties | (Ellis et al. 2018) |
| Host country's fractionalization | (Ellis et al. 2018) |

### 4.1.1. Cross-National Distance

Various cross-national distances affect EMNEs' choice of entry modes. Institutions are interpreted as the rules of the game in society, and institutional distance is an indicator of the degree of difference between countries (Kedia and Bilgili 2015). In general, a larger institutional distance often relates to a higher level of uncertainty, thus EMNEs as foreign investors are less likely to enter through the establishment of wholly-owned subsidiaries (WOS) and prefer international joint ventures (IJV) (Wu et al. 2021; Malhotra et al. 2016; Kedia and Bilgili 2015; Pinto et al. 2017; Kittilaksanawong 2017; Scalera et al. 2020). Regulation, norm, and cognition are the three pillars of institutional environments. Many researchers proposed that the greater the regulative distance, normative distance, and cognitive distance, the lower the preference of EMNEs for WOS and higher preference for IJV (Ilhan-Nas et al. 2018; Liou et al. 2017b).

Additionally, institutional distance can be further divided into formal institutional distance and informal institutional distance. The formal institutional distance, such as differences in politics, regulations, and law enforcement between countries, elevates EMNEs' chances of choosing WOS rather than IJV (Wu et al. 2021). In contrast, the informal institutional distance, such as differences in codes of conduct, norms, values and group behaviors among countries, reduces EMNEs' preference of WOS and raise the probability of adopting IJV (Liou et al. 2016; Ellis et al. 2018; Liou et al. 2017b). Nevertheless, after comprehensively considering the impact of formal and informal institutions on corporate decision-making and behaviors, some scholars proposed that the likelihood of establishing WOS increases when the institutional distance between the home and host countries is high (Lahiri et al. 2014; Yang 2015).

Moreover, culture varies from region to region, and cultural distance is reflected in different norms and behaviors (Malhotra et al. 2011). Many scholars pointed out that national cultural distance impacts the internationalization measures of EMNEs, including the choice of entry mode (Malhotra et al. 2011). Malhotra et al. (2011) believed that there is a curvilinear relationship (U shape) between cultural distance and the probability of choosing WOS as the entry mode. The probability of choosing WOS first drops as the cultural distance increases; in contrast, after a certain point, the relationship between national culture distance and the likelihood of establishing WOS becomes positive (Malhotra et al. 2011). However, most scholars stated a linear relationship that with a high level of cultural distance between the host and home countries, EMNEs are more likely to consider IJV over WOS (Malhotra et al. 2016; Yang 2015; Pan 1996; Demirbag et al. 2009; Kim et al. 2020; Tseng and Lee 2010). Likewise, when the linguistic and religious distance between the home country of the foreign entrants and the target country is significant, EMNEs tend to seek building IJV rather than full ownership structure in the venture (Dow et al. 2016; Demirbag et al. 2009).

Furthermore, impacts from geographic and psychic distance on EMNEs' entry mode choice have also been discussed. Malhotra and Gaur (2014) found a U-shaped relationship between geographic distance and the likelihood that EMNEs choose WOS. They posed a negative relationship between geographic distance and the likelihood of choosing WOS when geographic distance is low to moderate, while a positive relationship between geographic distance and the probability of choosing WOS when geographic distance is high (Malhotra and Gaur 2014). The impacts from psychic distance on EMNEs' entry mode choice are contingent on the features of host countries. Specifically, for EMNEs internationalizing to other emerging economies, the larger the psychic distance, the lower the rate of establishing WOS in the host country. While for EMNEs internationalizing to developed economies, the larger the psychic distance, the greater the probability of forming WOS instead of IJV in the host nation (Chikhouni et al. 2017).

### 4.1.2. Cultural Attributes

Cultural attributes of the target country and the home country also affect the EMNEs' entry options. The cultural attributes discussed in the selected papers include the power

distance of EMNEs' home countries, linguistic diversity, and religious diversity in host countries. Power distance represents the formal hierarchy of power in a society. Specifically, in a society with greater power distance, top-down decision-making and power inequality are universal and generally accepted (Hofstede 1980). Additionally, EMNEs from large power distance countries with strong capabilities have the resources to absorb and deal with the higher costs of establishing WOS. Thus, they are more likely to adopt a WOS rather than an IJV mode in the context of international business (Richards and Yang 2007). Moreover, similar to the fact that linguistic and religious distances between countries increase the difficulty of communicating and understanding with people of another nation, the existence of diversified languages and religions in the target regions also has an adverse effect on collecting market information (Dow et al. 2016). Dealing with multiple new languages and religions leads to the risk of misunderstanding and errors in cross-border transactions; thus, EMNEs prefer IJV over WOS as the entry mode, as they can learn from their local partners (Dow et al. 2016).

### 4.1.3. Government-Related Factors

Many government-related factors of home and host countries have important impacts on MENEs' entry model choice. Those factors cover a broad range, including the relationship between countries, media freedom, economic policies, governance effectiveness and regulations, and stability.

The relation between the home country and the host country impacts how EMNEs choose their entry modes. Over time, the changing strategic salience between the home country and the host country affects EMNEs' market entry choice (Zhu and Sardana 2020). For example, when there is an amiable commercial and diplomatic relation, EMNEs are more likely to choose WOS over IJV (Demirbag et al. 2010).

Various risks in host countries perceived by EMNEs, such as environmental uncertainty, regulatory unpredictability, and political hazards, can reduce the probability of choosing WOS over IJV. The environmental uncertainty faced by EMNEs is usually due to emergencies in the political, social and economic environment (Richards and Yang 2007). A country with high regulatory unpredictability usually exhibits frequent and unforeseen changes in government policies and has insufficient mechanisms for enforcing laws and contracts (Tseng and Lee 2010). When the host country has high environmental uncertainty, it is essential to maintain local partnerships to gain more local operational and management information (Bretas et al. 2021). Therefore, when the host country has higher environmental uncertainty, EMNEs are less likely to choose WOS over IJV as the entry mode (Richards and Yang 2007; Bretas et al. 2021). Under this circumstance, foreign entrants may receive less preferential treatment in many cases, making them less willing to establish WOS and prefer intermediate ownership cooperation (Tseng and Lee 2010; Pan et al. 2014). Moreover, political hazard refers to political instability, unforeseen changes in regulations, the probability of the host government intervention or disruption of cross-border investments, and vague laws and regulations (Demirbag et al. 2009). When EMNEs believe that their arrangements are at risk due to the weakness of the host country's political environment, they will tend to opt for lower equity participation and select a lower commitment entry mode (such as IJV) compared to a WOS (Moschieri et al. 2014; Lu et al. 2018; Demirbag et al. 2009; Demirbag et al. 2010).

According to Luo (2001), environmental dynamism, heterogeneity, and hostility decrease EMNEs' willingness to choose WOS while preferring IJV as an entry mode. The vitality of the host country involves the unpredictability and variability of environmental elements, which will increase the transaction costs, investment conversion and exit costs of foreign and local companies. Therefore, EMNEs will be less proactive in a changing environment (Luo 2001). Furthermore, the complexity of the environment increases the incidents that foreign companies need to deal with and brings more difficulties for them to adapt to environmental emergencies, make strategic decisions, and deploy production resources internationally (Luo 2001). Moreover, environmental hostility brings contractual

hazards for foreign companies, such as risks in resource procurement and infrastructure access (Luo 2001). As a result, EMNEs incline to choose WOS to minimize the transaction costs.

Further, a high degree of digital media freedom in the target country positively impacts the probability of EMNES choosing WOS as the entry mode for foreign investment. A high degree of freedom of digital media improves the transparency and openness of information flow and enables the opinions of different stakeholders to be published publicly without fear of retaliation (Burgess 2010). As latecomers to global competition, EMNEs are usually negatively affected by the country-of-origin effects in cross-border investment (Lahiri et al. 2014). By using free digital media, EMNEs can communicate directly with stakeholders, reducing the importance of local partners in alleviating the host country's information asymmetry (Li et al. 2021). Consequently, WOS is preferred by EMNEs.

Moreover, government incentives and restrictions for foreign investors lead to different ownership structures of multinational firms (Pan 1996). In some emerging economies such as China, local authorities may force foreign entrants to have an IJV arrangement over full ownership (Pan 1996). The authorities of state-led economies have certain political influence and bargaining power over foreign investment, and the local authorities are worried about losing control by giving cross-border companies full ownership (Chen 1994). Thus, local enterprises with a high level of state ownership tend to conduct intermediated ownership deals with multinational firms (Pan 1996), resulting in an IJV arrangement.

The host country's market environment is one of the key considerations for enterprises to consider FDI strategies (Han et al. 2021). The host country's market potential and market orientation also affect the choice of EMNEs' entry mode. Most scholars believed that a high level of market potential and market orientation of the host country increases the probability of foreign entrants choosing WOS (Nguyen and Binh 2021). In contrast, the high export orientation of the host country reduces the probability of EMNEs choosing WOS (Nguyen and Binh 2021). As a country's market orientation makes its government pay more attention to the domestic economy, its economic development will be more prosperous. Therefore, EMNEs can obtain more development opportunities in the country while the host country's export orientation increases its investment in overseas projects, thereby reducing the development opportunities of multinational companies in the country (Nguyen and Binh 2021). Additionally, Malhotra et al. (2011) stated that the influence of the target country's market potential on EMNEs' entry mode follows an inverted-U shape, where the probability of WOS increases as the market potential raises to a certain point and declines thereafter. In regard to market orientation, developed countries generally have more established distribution channels and business networks; thus, companies face less regulatory intervention. While the markets in emerging economies tend to be more unpredictable and unstable (Luo 2001). Hence, the probability of EMNEs choosing WOS as entry mode is positively associated with the target country's market orientation and negatively correlated with its export orientation (Luo 2001).

Government effectiveness can increase the willingness of EMNEs to select WOS instead of IJV, and there is a positive relationship between the rule of law, corruption control and the probability of WOS entry mode (Lahiri 2017). Similarly, the uncertainty of law and order and the pervasiveness of corruption reduce EMNEs' willingness to establish WOS but prefer IJV (Demirbag et al. 2010). In addition, Driffield et al. (2014) believed that the worse the government effectiveness of the host country, the lower the uncertainty of law and order, and the higher the degree of corruption, the higher the preference of EMNEs to pursue WOS over IJV. In a favorable foreign institutional environment, such as countries with a high degree of economic freedom index, multinationals are more inclined to choose a full ownership entry mode and less adopt IJV (Pan et al. 2014). Furthermore, the higher the degree of generalized institutional inefficiencies in the host country, the lower the probability of EMNEs entering through WOS than IJV. In contrast, the higher the degree of arbitrary institutional inefficiency in the host countries, the more likely the companies are to pursue WOS rather than intermediate ownership deals (Falaster et al.

2021). Moreover, legal requirements of the host country are negatively related to WOS against IJV (Mjoen and Tallman 1997). The greater the strength of intellectual property institutions in host countries, the more likely it is for EMNEs to undertake full acquisition rather than IJV (Ahammad et al. 2018).

Just like the host country's regulatory restrictions on inward FDI increase the trend of EMNEs choosing WOS instead of IJV, regulatory restrictions from the EMNEs' home country enhance the firms' preference for WOS instead of IJV (Cui and Jiang 2012), especially considering the state-owned EMNEs (Zhou 2018). Besides, support from the home country government makes up the EMNEs' disadvantages of ownership and their lack of internationalization experiences (Han et al. 2018), and prompts emerging companies to choose WOS instead of intermediate ownership plans in cross-national business (Pinto et al. 2017). While the pressure of the home country government suppresses firms' choice to establish WOS (Chung et al. 2016).

Furthermore, existing literature has conducted much research on the foreign investment mode of EMNEs on WOS vs. IJV, while the discussion of cross border acquisition (CBA) vs. greenfield investment is still in the exploratory stage based on the result of this review. The faster the development of formal institutions in host countries, the less likely it is for EMNEs to enter through an acquisition mode rather than a greenfield mode. Further, the faster the factor market of the host country develops, the more likely it is for EMNEs to enter through an acquisition mode instead of a greenfield mode (Chen et al. 2017). Meanwhile, based on the available research taking CBA vs. greenfield as the dependent variable, EMNEs are more likely to choose greenfield investments than acquisitions when they are located in origin clusters of foreign economies. However, when they are located in more advanced industry clusters of foreign economies, they are more likely to pursue CBA than greenfield (Shen and Puig 2018).

### 4.1.4. Historical and Social Factors

As mentioned above, the factors at the country-level that affect the EMNEs' entry mode typically include the distance between countries, cultural characteristics, government supervision and policies. In addition to these three main aspects, some historical and social factors, such as colonial history and fractionalization in host countries, affect how EMNEs enter the host countries.

In many colonies, such as African countries, foreign entrants exploited resources and conquered indigenous peoples with little regard for economic and institutional development. This kind of exploitation not only creates institutional voids in the system but also has a lasting psychological impact on many colonial regions' peoples (Athow and Blanton 2002; Jack et al. 2011; Mizuno and Okazawa 2009). Consequently, the past colonial history influences target firms' willingness to trust and cooperate with foreign acquirers and has a negative impact on EMNEs' establishment of WOS (Ellis et al. 2018).

A high degree of ethnic and language segmentation is the key feature of a fractionalized country (Ellis et al. 2018). The economies of fractionalized countries are often unstable because they generally lack a unified and consistent economic policy. This ambiguity in the availability of local resources and the demand creates uncertainty in the scope and timing of the resource investment required for foreign acquirers (Ellis et al. 2018). Hence, EMNEs prefer to use WOS rather than IJV when entering highly fractionalized countries to obtain the strategic control power required for management fractionalization and minimize the transaction costs involved (Ellis et al. 2018).

### 4.2. Industry-Level Factors

In addition to the environmental factors at the country level, the environmental factors at the industry level also play a decisive role in EMNEs' choice of entry mode. This is because the industry environment affects EMNEs' strategy significantly (Choi et al. 2020). The 44 papers we reviewed mainly examined EMNEs from the manufacturing and service industries. The industry-level environmental factors are divided into four main areas:

industry conditions, financial indicators at the industry level, industrial characteristics and trends, and technology (see Table 4).

**Table 4.** Industry-level environmental volatility factors that affect EMNEs' entry mode choice.

| Independent Variables | Reference |
|---|---|
| *Industry conditions* | |
| Competitive intensity | (Pan 1996) |
| Industry relatedness | (Ahammad et al. 2018; Yang 2015) |
| Market turbulence | (Tseng and Lee 2010) |
| *Financial indicators at industry-level* | |
| Local purchasing for production inputs | (Yu et al. 2015) |
| Local market for its sales | (Yu et al. 2015) |
| Large capital requirements | (Nguyen and Binh 2021) |
| *Industrial characteristics and trends* | |
| Industry experiences by peer firms | (Xie and Li 2017) |
| Imitation power | (Yang and Hyland 2012) |
| *Technological level of domestic industries* | |
| Home country industry technological capabilities | (Wang et al. 2019) |

### 4.2.1. Industry Conditions

Compared to their DMNEs' counterparts, as latecomers in the international market, EMNEs are relatively small in scale, lack international technological and managerial systems, and do not have international prestige and reputation (Tseng and Lee 2010). However, with the increasing industry globalization, EMNEs are frequently involved in global expansion and competition (Cui et al. 2017). Therefore, when facing a competitive target market, EMNEs are more inclined to choose IJV with a lower degree of risk than WOS (Pan 1996).

Industry relatedness is characterized by the difference or distance between the foreign investors and the target company in the industry (Ahammad et al. 2018). The degree of familiarity with the target industry greatly affects the entry mode decision of multinational firms (Morosini et al. 1998; Yin and Shanley 2008). High industry relatedness helps EMNEs absorb knowledge more quickly and develop entry strategies more efficiently (Ahammad et al. 2018; Yang 2015). Furthermore, benefiting from the familiarity with the target industry, EMNEs are unlikely to fall victim to the opportunistic behaviors of the potential partners in the target market (Malhotra et al. 2011). Thus, EMNEs are more likely to pursue WOS than IJV considering the lower risks due to the familiarity of the target industry. In contrast, when the foreign investors and the potential partners are in different industries, it is challenging to measure and orchestrate each other's technical know-how, managerial experiences and resources, which results in integration and valuation difficulties (Yang 2015). Therefore, EMNEs prefer IJV instead of WOS.

Market turbulence indicates the trend and degree of changes in customers, competitors, and business opportunities in certain industries (Tseng and Lee 2010). A predominantly turbulent target country market is full of interference for multinational companies to obtain information and knowledge to identify competitive behaviors and future development trends (Lee et al. 2008). Even for large multinationals from developed countries, agitated markets are full of challenges, not to mention emerging multinational enterprises, which are relatively smaller in scale and less experienced in cross-border investment (Francis et al. 2009; Tseng and Lee 2010). A large number of empirical studies show that to quickly and deeply understand and adapt to the market conditions of the host country, EMNEs tend to establish joint ventures to cooperate with local companies (Tseng and Lee 2010). This close interaction between partners helps EMNEs understand partners' business practices and makes it easier to establish contacts with host countries (Chen 2006). Thus, the greater the extent of market turbulence EMNEs perceive in the host country, the more likely for them to prefer IJV rather than WOS (Tseng and Lee 2010).

### 4.2.2. Financial Indicators at Industry Level

Local purchasing for production inputs plays a vital role in EMNEs' overseas expansion. Many EMNEs enter foreign countries to obtain resources not available in their own country or at higher costs than the target markets (Nachum and Zaheer 2005). Furthermore, most EMNEs give priority to the source of supply because their core value comes from material procurement and production (Yu et al. 2015). Consequently, establishing close cooperative relations with local companies can ensure the supply of raw materials and significantly improve the efficiency of EMNEs, taking into account local partners' knowledge, experience, and network (Yu et al. 2015). Therefore, compared with setting up WOS, when local procurement and production inputs are pivotal in the industry, EMNEs are more likely to cooperate with local partners through IJV (Yu et al. 2015).

Another crucial motive for entering foreign markets for EMNEs is to seek markets (Nachum and Zaheer 2005). When the primary value comes from downstream activities such as marketing and sales, clearly identifying and constantly updating the needs of local customers and managing the relationship with them is essential (Yu et al. 2015). Multinational companies tend to position themselves close to actual and potential customers to learn more about local consumers' specific demands and needs through more regular interactions (Yu et al. 2015). Additionally, due to local embeddedness and weak network connections, it is often more difficult for cross-national firms to obtain local information and resources, resulting in inefficiency (Zaheer 2002). Moreover, as the physical nature of the distribution system and the problem of international brand transfer, downstream activities involve fewer cross-border transfers of resources and capabilities compared with upstream activities (Yu et al. 2015). In contrast, local companies often have better local knowledge and social resources and are in an advantageous position in local business networks (Yu et al. 2015). Additionally, multinationals need to establish social relationships with different local market participants to develop resources and capabilities related to downstream activities in the host country's environment (Yu et al. 2015). Therefore, when the local market for sales in the industry is important, EMNEs are more likely to choose an IJV to a WOS when entering foreign markets (Yu et al. 2015).

The scale of investment and required funds differ across industries (Nguyen and Binh 2021). Multinational investors are very cautious about transactions involving large investment commitments because huge investments are accompanied by higher costs and risks (Nguyen and Binh 2021). Additionally, when a cross-national enterprise is engaged in an industry that requires a large amount of capital investment, the financial resource commitment of the local partners is vital (Wei et al. 2005). Therefore, in an industry that requires enormous investment, EMNEs tend to share risks with local partners and choose IJV instead of WOS (Nguyen and Binh 2021). In contrast, WOS is the preferred option when the investment is small because it allows EMNEs to fully control their companies and make profits (Shieh and Wu 2012).

### 4.2.3. Industrial Characteristics and Trends

In international trade, many companies have cross-organizational imitation in their entry methods, that is, a process in which individual firms are affected by peer companies (Henisz and Delios 2001; Phillips and Zuckerman 2001; Xie and Li 2017; Yang and Hyland 2012). When one or more enterprises adopt a strategic entry mode, it increases the probability of other companies adopting similar strategies (Henisz and Delios 2001). Many scholars believe that by learning from peers' experience, imitators are more likely to enjoy social or technical benefits (Yang and Hyland 2012). Due to the relative lack of global experience, management capabilities and expertise, EMNEs usually face greater uncertainty than DMNEs; thus, they tend to imitate industry experiences by peer firms when making entry mode decisions (Henisz and Delios 2001; Phillips and Zuckerman 2001).

For EMNEs, there are two crucial imitation targets, one is compatriot enterprises, and the other is DMNEs (Xie and Li 2017). When firms consider investing in another country, they have to observe how peer companies from their home country invested in the host

countries (Yang 2015). The decisions of peer companies are informative to the local company due to the same developing path in an emerging economy (Yang and Hyland 2012). Therefore, the more frequently the entry mode previously selected by peers in the host country, the greater the tendency of EMNEs to use the same entry mode (Xie and Li 2017). In addition, DMNEs are another prominent imitation object for EMNEs (Xie and Li 2017). Because DMNEs, accounting for more than 80% of the world's investment each year, are generally considered successful and owning rich international experience, competitive advantages, high status and legitimacy (Xie and Li 2017). Hence, the more frequently DMNEs adopt the entry mode in a certain environment, the greater the tendency of EMNEs to use the same entry mode when entering the country (Xie and Li 2017).

### 4.2.4. Technological Level of Domestic Industries

The technological level of an industry affects the entry mode choice of enterprises when making cross-border investments (Salomon and Jin 2008; Wang 2009). Due to differences in histories and natural endowments, the overall technological level of different industries in the same country differs significantly (Salomon and Jin 2008). In emerging economies, as local governments need to focus their limited financial and material support on characteristic pillar industries, one or more industries usually have technological leadership, while the rest are still underdeveloped in terms of technologies (Salomon and Jin 2008). Additionally, when entering foreign markets, EMNEs with powerful industry-level technologies can transform these country-specific advantages into their own company-specific advantages (Wang 2009). For example, benefiting from the huge scale of the domestic market and long-term support from home governments, Chinese nuclear energy companies need less host country technical resources for international export trade and tend to increase the use of resource development entry modes (Wang 2009). Therefore, with the improvement of domestic industrial technology capabilities, emerging market companies tend to prefer WOS rather than joint ventures when entering foreign markets (Wang 2009).

## 5. Discussion and Future Research Direction

As emerging economies become more critical and diversified, research on the determinants of EMNEs' entry mode has gradually increased; however, there are still some significant research gaps that provide promising directions for future research. We identified three research gaps and called for further investigation based on our systematic review of existing research (see Table 5).

**Table 5.** Future Research Direction.

| Research Gaps in Extant Studies | Reasons | Future Research Direction | Drivers | Future Prospect Offered |
| --- | --- | --- | --- | --- |
| Lack of research on external environmental factors | 1. Scholars mainly focused on DMNEs in the past because DMNEs dominated the international market. 2. The external environment was relatively stable in the past decades. | Varieties of environmental volatility factors | 1. EMNEs play an increasingly important role in the global economy. 2. The international environment has become complex and turbulent in recent years. | 1. Longitudinal research provides valuable opportunities to explore some exciting research questions. 2. Research on dynamic environmental volatility factors provides a deeper understanding of the influences. 3. Makes the studies more dynamic and comprehensive, facilitating the development of extant theories. |

**Table 5.** *Cont.*

| Research Gaps in Extant Studies | Reasons | Future Research Direction | Drivers | Future Prospect Offered |
|---|---|---|---|---|
| Lack of research on EMNEs from less concerning emerging economies | The internationalization of EMNEs from dominant emerging economies (e.g., BRICs) is more proactive and taking an increasing share in international business. | Varieties of emerging economies | Environmental factor changes in the volatile circumstances will have different impacts on different economies. | 1. Helps scholars to further understand the similarities and differences of the relationship between the environmental factors and EMNEs' entry mode choice in different economies. 2. Help scholars better understand the boundary conditions of environmental factors affect EMNEs' entry mode decision-making under volatile circumstances. |
| Lack of research on SMEs | 1.Large EMNEs are the dominant participants in international business. 2. The data of listed companies are easier to obtain. | Varieties of companies' scale | 1. SMEs are expected to have heterogeneous risk preferences, impacting entry mode choice. 2. SMEs might be more flexible under a new environment. 3. The international expansion of SMEs from emerging economies has become an important phenomenon in the international market. 4. Born global firms as a specific category of SMEs that deserves more attention. | 1. Help scholars understand the similarities and differences of the environmental drivers on enterprises of different sizes. 2. Enables researchers to extend relevant theories regarding environmental factors that affect SMEs' entry mode choice under turbulent circumstances. 3. Provides scholars with great opportunities to develop theories which might be more suitable for the internationalization of SMEs from emerging economies. |

### 5.1. Varieties of Environmental Volatility Factors

It can be seen from the final selection of only 44 articles in line with the selection criteria that the research on the determinants of EMNEs' entry mode with environmental issues is relatively limited. The reasons are twofold. First, scholars mainly focused on DMNEs in the past because DMNEs dominated the international market. Economies such as the U.S. and Japan, have gained the primary attention of many scholars. For example, Lo (2016) studied the American multinationals internationalization strategy, while Chen and Hennart (2004) studied the Japanese companies. However, as EMNEs play an increasingly important role in the global economy, a growing number of scholars began to study EMNEs.

Second, the external environment was relatively stable in the past two decades. Thus, scholars paid more attention to the firm-level factors other than external environmental factors when studied on EMNEs (Li and Meyer 2009; Gubbi 2015; Gaffney et al. 2016). However, the international environment has become complex and turbulent in recent years. For instance, the Sino-U.S. trade war has been challenging for EMNEs, and the COVID-19 outbreak further aggravates the turbulence of the international environment. The volatile environment affects environmental factors at both country- and industry-levels; thus, it has a significant impact on EMNEs' internationalization strategies and entry mode decision-making. Therefore, it is important to study the diverse EMNEs' environmental factors.

We identified both the country-level and the industry-level factors and classified them into four different categories, respectively, in Section 4. However, as Tables 3 and 4 show, compared with some environmental factors that have been discussed many times, such as institutional distance and cultural distance under cross-national distance dimension and

host country risk under government-related factors dimension at the country-level, some environmental factors lack the attention of scholars. For example, the studies involved the factors of geographic distance and psychic distance under cross-national distance dimension are limited. In addition, industry-level factors did not receive enough attention in extant studies.

Moreover, investigating the time dimension extends our knowledge about EMNEs' entry mode choice. Longitudinal research provides valuable opportunities to explore some exciting research questions. For example, it might be inspiring to explore environmental factors' effects on the sequence of entry and mode switch (Zhao et al. 2004). In addition, environmental factors are dynamic in their nature. Examining impacts from the change of environmental factors provides a deeper understanding of the influences from environmental factors.

Therefore, more research is needed for these environmental factors that are less investigated or even not mentioned in the selected 44 articles. The diversification of environmental factors can make the studies on the determinants of EMNEs' entry mode more comprehensive, facilitating the development of extant theories.

### 5.2. Varieties of Emerging Economies

The limited studies on EMNEs' decision-making with volatile environmental issues reviewed in this paper cover restricted emerging economies as EMNEs' home countries. These studies mainly focus on the BRIC, Turkey, Mexico, and Southeast Asia, especially China, which occupied an absolute position. The main reason is that the internationalization of EMNEs from these economies is more proactive and taking an increasing share in international business, thereby grabbing the attention of scholars.

However, focusing only on these dominant economies is not enough, and more future research is needed for other emerging economies as well, such as CEE, Latin America and UAE. This is because environmental factor changes in the volatile circumstances will have different impacts on different economies. Therefore, EMNEs from different emerging economies need to consider the environmental factors differently. For example, in the post-COVID-19 era, China and CEE economies face different social and economic realities. Chinese multinationals might consider the environmental factors such as the host country's risk, while companies from CEE might lay more emphasis on environmental factors related to home country risks. Thus, investigating EMNEs from other emerging economies provides opportunities for comparative research, which is needed to advance relevant theories (Alon et al. 2018).

In sum, we call for more insight into the studies on diverse emerging economies in future studies. It helps scholars to further understand the similarities and differences of the relationship between the environmental factors and EMNEs' entry mode choice in different economies. Moreover, studying diverse emerging economies can help scholars better understand the boundary conditions of environmental factors that affect EMNEs' entry mode decision-making under volatile circumstances.

### 5.3. Varieties of Companies' Scale

Many extant studies investigated large EMNEs, especially listed companies, but few in SMEs. This is because, on the one hand, large EMNEs are the dominant participants in international business; on the other hand, the data of listed companies are easier to obtain than that of SMEs.

Theoretically, firm size is an important moderator during the organizational decision-making process (Haddoud et al. 2021). SMEs are limited in resources to deal with the uncertainty that occurs in the internationalization process (Holtgrave and Onay 2017). Thus, SMEs are expected to have heterogeneous risk preferences compared with large firms, impacting their entry mode choice. They also face difficulties to build connections with firms in host countries due to a lack of reputation (De Maeseneire and Claeys 2012). In addition, SMEs face more institutional challenges, such as power imbalance, when

cooperating with local enterprises with large scale differences (Terpend and Ashenbaum 2012). On the other hand, SMEs might be more flexible to relocate and redeploy their capabilities and resources into a new environment (Neirotti and Raguseo 2017).

Methodologically, investigating EMNEs with varied sizes helps alleviate the large-firm bias (Zhao et al. 2004). Practically, the international expansion of SMEs from emerging economies has become an important phenomenon in the international market (Qiao et al. 2020). Therefore, more research on SMEs is needed in the future.

Moreover, born global firms (BGs) is one specific category of SMEs that deserves more attention. BGs are typically small-sized and young with limited internal resources (Dzikowski 2018). Meanwhile, BGs are key participants in the global market that cannot be neglected. BGs' demand for outward FDI is innate, and they undertake international business from the earliest days of the establishment (Madsen and Servais 1997). Zahra and George (2017) argued that the changing environmental drivers are highly likely to foster continued internationalization. From a short-term perspective especially, environmental factors were found to be influential on BGs' internationalization strategies (Øyna et al. 2018). Thus, those companies may be more sensitive to environmental turbulence and need to be aware of environmental changes in advance (Holtgrave and Onay 2017).

Therefore, the research on the environmental factors of not only large companies but also SMEs is becoming increasingly important. Research on multinational firms of different sizes in emerging economies can expand the diversification of extant research. It will help scholars understand the similarities and differences of the environmental drivers on enterprises of different sizes. In addition, it enables researchers to extend relevant theories regarding environmental factors that affect SMEs' entry mode choice under turbulent circumstances. Moreover, it might even provide scholars with great opportunities to develop theories which might be more suitable for the internationalization of SMEs from emerging economies.

## 6. Conclusions

Motivated by the importance of the entry mode choice of EMNEs when entering a new market, this study reviewed and analyzed 44 selected peer-reviewed articles published from 1996 to June 2021, and identified what and how volatile environmental factors influence EMNEs' decision-making on foreign entry mode choice.

Our review identified three research gaps and provided future research directions to scholars. (1) More research on external environmental volatility factors is needed when exploring influencing factors toward EMNEs' entry mode choice. (2) Scholars shall pay more attention to EMNEs from countries other than large emerging economies. (3) More studies are needed on SMEs among EMNEs, but not only listed firms.

Practically, EMNEs need to understand the volatile environmental factors because the decision-making basis is to enhance the companies' "competitiveness, efficiency and control over critical resources" (Yiu and Makino 2002, p. 667). An in-depth understanding of environmental factors enables an EMNE to better analyze its own situation in a turbulent environment and formulate the most suitable strategy according to the needs and objective conditions, thereby selecting the most competitive entry mode. Our review, therefore, provides EMNEs with the practical implication that environmental factors play a vital role in helping them choose appropriate entry modes. Hence, EMNEs should pay attention to those factors in evaluating different entry mode choice, especially under volatile environments.

We recognize two major limitations of this review research. Firstly, the current review focuses on the equity-based entry mode, which excludes non-equity modes of entry. Equity-based entry mode is the most widely investigated entry mode among the extant research. However, there are other dimensions of entry mode to explore, such as non-equity entry mode. As Ahsan and Musteen (2011) suggested, more attention is needed to inspect the non-equity modes of entry, such as exporting and licensing, especially when environmental volatility is high. This suggestion applies to EMNEs, which are more sensitive to the

environmental factors during the decision-making process of internationalization. A better understanding towards non-equity entry modes increases our knowledge about ENMEs internationalization strategies and provides insights for EMNEs' managers when making entry mode choice.

Secondly, the current review does not provide a detailed discussion about the theoretical perspectives adopted in extant literature. We briefly summarized the theories and perspectives used in the selected journal articles in Table 2 without discussion in details. EMNEs' internationalization process offers opportunities to develop extant theories as EMNEs are from developing economies with heterogenous social, institutional, and economic nature (Luo et al. 2019). A more in-depth review of the theoretical perspectives provides scholars with knowledge about existing mechanisms to explain antecedents of EMNEs' entry mode choice and points out potential directions to further advance theories to better explain this phenomenon.

**Author Contributions:** Conceptualization, Y.L. and R.G.; methodology, Y.L. and R.G.; investigation, Y.L. and R.G.; resources, Y.L., R.G. and J.W.; writing—original draft preparation, Y.L. and R.G.; writing—review and editing, Y.L., R.G. and J.W.; visualization, Y.L., R.G. and J.W.; supervision, Y.L.; project administration, Y.L.; funding acquisition, J.W. All authors have read and agreed to the published version of the manuscript.

**Funding:** The authors gratefully acknowledge the financial support from the School of Management and Governance, UNSW Business School.

**Acknowledgments:** All authors gratefully acknowledge support from Yi Li. All authors would like to thank him for his careful supervision and comments, which greatly improved the paper. All authors are grateful to editors for their time and assistance on relevant work of manuscript submission, revision, acceptance, proofreading and publication. Additionally, we strongly appreciate reviewers who provide constructive suggestions that inspire our ideas and improve the article.

**Conflicts of Interest:** The authors declare no conflict of interest.

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
