# Peer review of "Determinants of EMNEs’ Entry Mode Decision with Environmental Volatility Issues: A Review and Research Agenda"

_jrfm, doi:10.3390/jrfm14100500_

Round 1

Reviewer 1 Report

Review of the paper titled „Determinants of EMNEs’ Entry Mode Decision with Environ-2 mental Volatility Issues: A Review and Research Agenda” under consideration to Journal of Risk and Financial Management:

  • At the end of the introduction (before the agenda) one could add a brief summary of the main points from the literature review.
  • Page 20, Line 646, get rid of a line (tracked changes mark on the left side of the page)
  • Page 20, Line 661, get rid of a line (tracked changes mark on the left side of the page)
  • Page 1, line 42 “U.S” –> US
  • Please make sure your references are following the journal standard
  • In conclusion, I would like to see the limitations of the study.

Author Response

Dear reviewers,

This document contains our responses to the review of Manuscript jrfm-1392871. We would like to thank you all for your comments and suggestions sincerely. Specific responses to your comments and suggestions are listed in the sections below.

Best regards,

Authors

Reviewer 2 Report

Dear Editor,

Thank you very much for offering me the opportunity to review this review analysis. The review analysis is interesting and well written. However, there is still room for specific narrative, organizational and argumentative improvements prior it to be accepted for publication. To this end my review comments can be considered.

Thank you very much again.

Kind regards,

Dr. Grigorios L. Kyriakopoulos

Review comments for the manuscript: jrfm-1392871

Determinants of EMNEs’ Entry Mode Decision with Environmental Volatility Issues: A Review and Research Agenda

The review analysis is interesting and well written. However, there is still room for specific narrative, organizational and argumentative improvements prior it to be accepted for publication. To this end the following review comments can be considered.

1) In the Abstract section the authors’ statement that “This review examines literature from 1996 to June 2021 on the determinants of EMNEs’ entry mode decision with environmental issues” can be accompanied by typical examples of specific “entry mode decisions” and specific “environmental issues” that are related to outward FDI for both EMNEs and SMEs, regionally and/or globally. Two or three extra sentences in the Abstract section are adequate.

2) Meticulous care should be taken that all research actions’ made in the past to be stated in past tense, no mixed present and past tense. Only an example of mixed present and past tenses (while only past tense is needed), but authors should check all other narrative, it is provided as follows:

“The remainder of the paper is organized as follows. Section 2 describes the research method used for conducting the review. Section 3 provides some descriptive statistics regarding bibliometric analysis, followed by a content analysis in Section 4, in which we classified the environmental factors as country-level factors and industry-level factors for detailed analysis. Next, in section 5, we proposed opportunities for future research. Finally, section 6 concludes the paper.”

3) It is not straightforward why in other headings authors are referred to “factors” and in other headings authors are referred to “variables” (in other text points there are also denoted as “indicators”). The “variables” are related to measurable entities of values’ taken and units’ measured that apply to equations and have undergone statistical modeling, while the “factors” can be descriptive entities. Therefore, conceptual differentiation among them and consistent notation of them throughout the narrative flow, Tables, Figures, they have to be denoted.

4) Subsection 4.1 is duplicated noted as “4.1 Country-level Variables” then follow the subsections 4.1.1 up to 4.1.4 and, after that another subsection 4.1 titled “4.1 Industry-level Variables”. The second-4.1 heading can change to 4.2.

5) Among the core-directions of the review objectives are that

  1. a) “The existing research mainly focuses on multinationals from large emerging countries, such as China and India”.
  2. b) “…..search criteria were set as follows: â‘´ search period: 1996 – June 2021”.

Nevertheless, while there are sporadic referencing points of time and regional-national value, these core-research objectives of: a) Regional factors and b) Chronological factors, they can be creatively developed in two distinct and separate sections or subsections, no in plain text that it is highlighting-focused on other issues. The regional factors can be allocated in specific geographical areas: continents, nations, or even in the vast geographical contexts of 1) Asia and 2) India. Besides, the chronological factors can be organized in the two time intervals of: 1996-2006, 2007-2021. In both these classifications: regional and chronological, the key-point is authors to determine what are the evolutionary features of the examined topic: from place to place, from time to time, or at the same place from time to time.

6) The content of the three subsections 5.1-5.3 has to be accompanied by a graphical representation or Table-formulation, in which the key-determinant per section, the drivers, the barriers, the constraints and the future prospects offered, they can be co-represented in graphical paths-links. This extra Figure, conceptual path, or Table has to note relevant source citations, in which authors considered while developing it in their own review analysis.

7) Due to the review content of the manuscript it is highly recommended authors to check a fresh and update literature production on the topic examined. To this end I made a literature search the outcome of which is presented below. Authors are recommended to enrich the theoretical and argumentative parts of their review analysis, accordingly.

Scopus

EXPORT DATE:23 Sep 2021

Soundararajan, V., Sahasranamam, S., Khan, Z., Jain, T.

57203952815;55949261300;56189972400;57150465500;

Multinational enterprises and the governance of sustainability practices in emerging market supply chains: An agile governance perspective

(2021) Journal of World Business, 56 (2), art. no. 101149, . Cited 4 times.

https://www.scopus.com/inward/record.uri?eid=2-s2.0-85092479264&doi=10.1016%2fj.jwb.2020.101149&partnerID=40&md5=3173514acb62a6c7d8ab6724f464e6c8

DOI: 10.1016/j.jwb.2020.101149

DOCUMENT TYPE: Article

PUBLICATION STAGE: Final

SOURCE: Scopus

Yang, Y., Lütge, C.

57209252637;22835631200;

Dynamic integration paths of emerging multinational enterprises in advanced markets: Empirical evidence from Chinese acquisitions in Germany

(2020) Review of International Business and Strategy, 30 (1), pp. 1-23. Cited 2 times.

https://www.scopus.com/inward/record.uri?eid=2-s2.0-85076597724&doi=10.1108%2fRIBS-05-2019-0052&partnerID=40&md5=e207a4a8b5b26225386cbe3f644600d9

DOI: 10.1108/RIBS-05-2019-0052

DOCUMENT TYPE: Article

PUBLICATION STAGE: Final

SOURCE: Scopus

Zhu, Y., Sardana, D.

57201342528;15729744700;

Multinational enterprises’ risk mitigation strategies in emerging markets: A political coalition perspective

(2020) Journal of World Business, 55 (2), art. no. 101044, . Cited 11 times.

https://www.scopus.com/inward/record.uri?eid=2-s2.0-85074231646&doi=10.1016%2fj.jwb.2019.101044&partnerID=40&md5=fd3349782d27ddeef9069de0ea256cb7

DOI: 10.1016/j.jwb.2019.101044

DOCUMENT TYPE: Article

PUBLICATION STAGE: Final

SOURCE: Scopus

Progunova, L.V, Trokhova, E.V, Milonova, M.V

57208470721;57195611414;57217158448;

Internationalization patterns of BRICS Multinational Enterprises MNEs: How differ from other emerging markets?

(2019) Espacios, 40 (35), art. no. 26, . Cited 3 times.

https://www.scopus.com/inward/record.uri?eid=2-s2.0-85089941550&partnerID=40&md5=4d245932d2a1af2e31dcce27cafe9b58

DOCUMENT TYPE: Article

PUBLICATION STAGE: Final

SOURCE: Scopus

….……………..

Han, X., Liu, X., Xia, T., Gao, L.

57196370342;35213572600;24466974100;55265988300;

Home-country government support, interstate relations and the subsidiary performance of emerging market multinational enterprises

(2018) Journal of Business Research, 93, pp. 160-172. Cited 18 times.

https://www.scopus.com/inward/record.uri?eid=2-s2.0-85046818892&doi=10.1016%2fj.jbusres.2018.04.021&partnerID=40&md5=408977a80b98b483c2b708ba0e46b077

DOI: 10.1016/j.jbusres.2018.04.021

DOCUMENT TYPE: Article

PUBLICATION STAGE: Final

OPEN ACCESS: All Open Access, Green

SOURCE: Scopus

Zhou, N.

57202363875;

Hybrid State-Owned Enterprises and Internationalization: Evidence from Emerging Market Multinationals

(2018) Management International Review, 58 (4), pp. 605-631. Cited 6 times.

https://www.scopus.com/inward/record.uri?eid=2-s2.0-85048033243&doi=10.1007%2fs11575-018-0357-z&partnerID=40&md5=5ced32fd41e846d54c8edede59c04290

DOI: 10.1007/s11575-018-0357-z

DOCUMENT TYPE: Article

PUBLICATION STAGE: Final

SOURCE: Scopus

Leposky, T., Arslan, A., Kontkanen, M.

57190009649;36619634600;57189999476;

Determinants of reverse marketing knowledge transfer potential from emerging market subsidiaries to multinational enterprises’ headquarters

(2017) Journal of Strategic Marketing, 25 (7), pp. 567-580. Cited 8 times.

https://www.scopus.com/inward/record.uri?eid=2-s2.0-84976345828&doi=10.1080%2f0965254X.2016.1195856&partnerID=40&md5=63259bc5a2e6bcb5f40f09db6c8d661f

DOI: 10.1080/0965254X.2016.1195856

DOCUMENT TYPE: Article

PUBLICATION STAGE: Final

OPEN ACCESS: All Open Access, Green

SOURCE: Scopus

Buckley, P.J., Tian, X.

7202909430;7202380213;

Internalization theory and the performance of emerging-market multinational enterprises

(2017) International Business Review, 26 (5), pp. 976-990. Cited 20 times.

https://www.scopus.com/inward/record.uri?eid=2-s2.0-85016006682&doi=10.1016%2fj.ibusrev.2017.03.005&partnerID=40&md5=8c764f298bfb8a5ebc0ebdfb6d803ee8

DOI: 10.1016/j.ibusrev.2017.03.005

DOCUMENT TYPE: Article

PUBLICATION STAGE: Final

OPEN ACCESS: All Open Access, Green

SOURCE: Scopus

Cui, L., Fan, D., Liu, X., Li, Y.

36522613900;35239542800;35213572600;57194342534;

Where to Seek Strategic Assets for Competitive Catch-up? A configurational study of emerging multinational enterprises expanding into foreign strategic factor markets

(2017) Organization Studies, 38 (8), pp. 1059-1083. Cited 44 times.

https://www.scopus.com/inward/record.uri?eid=2-s2.0-85026385893&doi=10.1177%2f0170840616670441&partnerID=40&md5=26119f3e66e7050a3927c69ddc3cd9cf

DOI: 10.1177/0170840616670441

DOCUMENT TYPE: Article

PUBLICATION STAGE: Final

OPEN ACCESS: All Open Access, Green

SOURCE: Scopus

Marano, V., Tashman, P., Kostova, T.

35345178800;35346183600;6603774111;

Escaping the iron cage: Liabilities of origin and CSR reporting of emerging market multinational enterprises

(2017) Journal of International Business Studies, 48 (3), pp. 386-408. Cited 163 times.

https://www.scopus.com/inward/record.uri?eid=2-s2.0-85018451498&doi=10.1057%2fjibs.2016.17&partnerID=40&md5=2a10bc3b2fb141a55bf74e0621f6a266

DOI: 10.1057/jibs.2016.17

DOCUMENT TYPE: Article

PUBLICATION STAGE: Final

SOURCE: Scopus

Author Response

(The authors gave the same response as above.)

Reviewer 3 Report

The review of the article provided made for an enjoyable read. The article is structured according to current standards. I appreciate the timeliness of the trends presented and the research results cited. The structure of the article was clearly and methodically developed. Aptly selected research procedure was used. In particular, I appreciate the clear presentation of the three identified gaps. 
Also noteworthy is the accuracy and timeliness of the selection of cited literature items of which as much as more than 40% are from the year 2017 and newer. 

For stylistic reasons, I would suggest reviewing the structure in terms of the so-called "orphans" at the end of lines which would increase the fluidity of reading as well as removing unnecessary spaces and gaps (e.g. in the bibliography). 
I also have one technical reservation. In the file provided to me, it seems that in section 2.3. the text breaks off. Is it supposed to be like that? Didn't the drawing obscure the rest of the text? Please verify and add/correct if necessary. 

Author Response

(The authors gave the same response as above.)

Round 2

Reviewer 2 Report

At this revised manuscript authors developed a substantially improved manuscript, having all review comments addressed in a systematic and careful manner. All research components have been creatively solidified, offering a coherent analysis of insightful remarks. Therefore, the revised manuscript sustains novel features and it can be accepted for publication at the “Journal of Risk and Financial Management” as is.

This manuscript is a resubmission of an earlier submission. The following is a list of the peer review reports and author responses from that submission.